

# Revolutionizing CAD/CAM-based restorative dental processes and materials with artificial intelligence: a concise narrative review

Hanin E. Yeslam[1], Nadine Freifrau von Maltzahn[2] and Hani M. Nassar[1]

[1] Department of Restorative Dentistry, King Abdulaziz University, Jeddah, Saudi Arabia
[2] Department of Prosthetic Dentistry and Biomedical Materials Science, Hannover Medical School, Hanover, Germany

## ABSTRACT

Artificial intelligence (AI) is increasingly prevalent in biomedical and industrial development, capturing the interest of dental professionals and patients. Its potential to improve the accuracy and speed of dental procedures is set to revolutionize dental care. The use of AI in computer-aided design/computer-aided manufacturing (CAD/CAM) within the restorative dental and material science fields offers numerous benefits, providing a new dimension to these practices. This study aims to provide a concise overview of the implementation of AI-powered technologies in CAD/CAM restorative dental procedures and materials. A comprehensive literature search was conducted using keywords from 2000 to 2023 to obtain pertinent information. This method was implemented to guarantee a thorough investigation of the subject matter. Keywords included; "Artificial Intelligence", "Machine Learning", "Neural Networks", "Virtual Reality", "Digital Dentistry", "CAD/CAM", and "Restorative Dentistry". Artificial intelligence in digital restorative dentistry has proven to be highly beneficial in various dental CAD/CAM applications. It helps in automating and incorporating esthetic factors, occlusal schemes, and previous practitioners' CAD choices in fabricating dental restorations. AI can also predict the debonding risk of CAD/CAM restorations and the compositional effects on the mechanical properties of its materials. Continuous enhancements are being made to overcome its limitations and open new possibilities for future developments in this field.

## INTRODUCTION

The use of computer-aided design/computer-aided manufacturing (CAD/CAM) technology in the construction of dental restorations has grown exponentially over the years since its introduction (*Duret, 1991*; *Spitznagel, Boldt & Gierthmuehlen, 2018*). Current dental CAD/CAM machines can mill different restorations using enhanced CAD software, digitizers, and scanners, with improved sensitivity (*Miyazaki & Hotta, 2011*). CAD/CAM dental procedures reduce the time and work required to produce dental

Corresponding author
Hanin E. Yeslam,
ayaslam@kau.edu.sa

restorations while saving the restoration data for later edits in design or restoration reproduction. Artificial intelligence (AI) plays a role in CAD/CAM dental systems. It helps incorporate esthetic factors, occlusal schemes, and previous practitioners' CAD choices in fabricating dental restorations (*Cho et al., 2024*; *Farook et al., 2020*; *Naidu & Jaju, 2022*). It can also predict the debonding risk of CAD/CAM restorations and the compositional effects on the mechanical properties of its materials (*Li et al., 2022*; *Yamaguchi et al., 2019*).

Using virtual reality (VR) simulation, pre-treatment esthetic restorative results can be provided in CAD/CAM machines (*Naidu & Jaju, 2022*). Machine learning (ML) can utilize collected *in vitro* mechanical test results combined with AI data analysis to predict the effect of a specific composite and PICN CAD/CAM material components on its flexural properties (*Li et al., 2022*). Unfortunately, most mechanical strength values of newly developed CAD/CAM materials are mostly acquired from manufacturers (*Grzebieluch et al., 2022*). For ML algorithms to accurately predict material properties, however, optimized hyperparameters and adequate datasets obtained from *in vitro* tests are important (*Li et al., 2022*; *Yang & Shami, 2020*). As early as 1995, the potential for AI to be utilized in CAD/CAM industrial engineering was proposed (*Ng & Poynting, 1995*). AI is being studied in dentistry, specifically in detecting lesions and identifying issues through radiographic analysis. There are some recent reviews that look into AI in various dental practices (*Alqutaibi & Aboalrejal, 2023*; *Ding et al., 2023b*; *Khanagar et al., 2022a*, *2022b*; *Ma et al., 2022*; *Mohammad-Rahimi et al., 2022*; *Mushtaq, Azam & Baig, 2022*; *Putra et al., 2022*; *Revilla-Leon et al., 2022*; *Schwendicke et al., 2022b*; *Talpur et al., 2022*). While AI and restorative dentistry area of research is relatively new, it is quickly advancing (*Alqutaibi & Aboalrejal, 2023*). This makes it challenging for dental practitioners to keep up with the latest and most useful developments. Those new to using AI in CAD/CAM and digital restorative dentistry may find it difficult to fully understand the various available applications. As far as the authors know, there is no updated review that concentrates on integrating dental CAD/CAM and AI. This concise narrative review discusses the integration of artificial intelligence in CAD/CAM restorative dentistry. It focuses on how AI can simplify restorative dental procedures, aid in material selection and development, dental restoration design, milling processes, and restorative prognosis.

## Relevance and target audience

Significant strides have been made in the realm of dental CAD/CAM technology, with various AI applications proving instrumental in driving progress. The objective of this review was to present a concise and simplified overview of the most recent and noteworthy dental CAD/CAM with AI developments rather than an exhaustive account of the existing literature. Thus providing valuable insights into the current state of available improvements for dental practitioners. To facilitate comprehension of these advancements among both experienced restorative dental specialists and newly graduated dentists who are newcomers to the field of AI in restorative CAD/CAM digital dentistry, a succinct overview of the capabilities and technologies that have resulted from the incorporation of AI in dental restorative CAD/CAM systems has been compiled.

## SURVEY METHODOLOGY

The search and narration in this article aimed to give readers with dental backgrounds a concise and simplified overview of the most recent dental CAD/CAM and AI advancements rather than a detailed report and appraisal of the available literature. Therefore, a bibliographic non-exhaustive search of the literature was conducted from January 2000 to January 2024. This is beneficial for dental practitioners seeking a timely update of the most recent advances or an introductory review of the subject in such a rapidly developing area of digital dentistry.

The databases used were Web of Science, PubMed/Medline, and Google Scholar. The combinations of the following keywords were used according to a free-text protocol, including: "Artificial Intelligence", "CAD/CAM", "Machine Learning", "Digital Dentistry", "Neural Networks", "Virtual Reality", and "Restorative Dentistry". The search only included articles written in English, excluding conference abstracts and commentaries. Articles mainly discussing industrial or medical CAD/CAM, and those related to other dental specialties were also excluded. A thorough search within article references and of recently advertised dental AI applications from various manufacturers was conducted to ensure comprehensive of the search, and the results were evaluated for inclusion carefully. The search only included articles written in English. The results were reported in a narrative format, without any quantitative data synthesis. The search revealed various artificial intelligence areas within digital restorative dentistry in general and CAD/CAM-specific current and potential AI applications.

The literature review was guided by the Preferred Reporting Items for Systematic Reviews and Meta-Analyses (PRISMA) search protocol, which emphasizes using its checklist and instructions for a literature review to capture the needed information (*Page et al., 2021*). The search strategy is detailed in Table 1.

### Artificial Intelligence and its pertinent branches

AI refers to machines performing human tasks using their own intelligence and a specific set of data. The goal is to enhance their abilities for quick, efficient, and accurate task completion (*Alauddin, Baharuddin & Ghazali, 2021*). It was first introduced in 1956 at Dartmouth University as computer-generated simulations of human cognitive abilities (*Obermeyer & Emanuel, 2016*). The latest advancements in AI, particularly the third generation linked with the fourth industrial revolution, are revolutionizing how dental and medical practices function (*Ding et al., 2023b*; *Thurzo et al., 2022b*). The integration of AI and data science has resulted in innovative improvements in dental healthcare services. In the field of AI, advanced machine learning on huge datasets (commonly known as "big data") is required to perform specific algorithms (*Alauddin, Baharuddin & Ghazali, 2021*; *Fatima et al., 2022*; *Rekow, 2020*).

AI can be categorized into three stages: weak (or narrow) AI, strong (or general) AI, and super (or genius) AI (*Ding et al., 2023b*; *Kulin et al., 2021*; *Thurzo et al., 2022b*). Narrow AI is designed to solve specific tasks and is the most common type of AI being used today, including applications such as reinforcement learning, face recognition, and data mining. Strong AI aims to have the same level of intelligence and awareness as humans and can

**Table 1 The detailed literature search strategy and methods.**

| Objective: | – The objective of this survey is to provide a concise and simplified overview of the most recent dental CAD/CAM and AI advancements for readers with dental backgrounds, rather than a detailed report and appraisal of the available literature. |
| --- | --- |
| | – The survey aims to conduct a non-exhaustive search of the literature from January 2000 to January 2024 and present the results in a narrative format without any quantitative data synthesis. |

**Search strategy:**

| 1. Databases: | – PubMed/Medline | – Web of Science | – Google Scholar |
| --- | --- | --- | --- |
| 2. Search terms:<br>-"Artificial INTELLIGENCE"<br>-"CAD/CAM"<br>-"Machine learning"<br>-"Digital dentistry"<br>-"Neural networks"<br>-"Virtual reality"<br>-"Restorative dentistry"<br>-"Natural language processing"<br>-"Fuzzy logic" | Search using Boolean operators:<br><br>("CAD/CAM" [MeSH] OR "digital dentistry" [MeSH] or "digital restorative dentistry" [MeSH] OR "computer aided design computer aided milling" [MeSH] or "restorative dentistry" AND ("Artificial intelligence" [MeSH] OR "Machine learning" OR "Virtual reallity" [MeSH] OR "AI-based" OR "Deep learning" [MeSH] OR "Fuzzy logic" [MeSH] OR "Natural language processing" [MeSH] OR "Neural networks" [MeSH])<br>Results (93 records) | ("CAD/CAM" OR "digital dentistry" OR "digital restorative dentistry" OR "computer aided design computer aided milling" OR "restorative dentistry") AND ("Artificial intelligence" OR "Machine learning" OR "Virtual reallity" OR "AI-based" OR "Fuzzy logic" OR "Natural language processing" OR "Neural networks" NOT (MEDLINE)<br>Results (159 records) | ("CAD/CAM" OR "digital dentistry" OR "digital restorative dentistry" OR "computer aided design computer aided milling" OR "restorative dentistry") AND ("Artificial intelligence" OR "Machine learning" OR "Virtual reallity" OR "AI-based" OR "Fuzzy logic" OR "Natural language processing" OR "Neural networks")<br>Results (203 records) |
| 3. Inclusion criteria: | – Articles written in English | | |
| | – Published between January 2000 and January 2024 | | |
| | – Relevant to restorative dental CAD/CAM and AI advancements | | |
| 4. Exclusion criteria: | – Conference proceedings, abstracts, and commentaries | | |
| | – Articles primarily discussing industrial or medical CAD/CAM | | |
| | – Articles related to other dental specialties | | |
| | – Articles related to restorative dentistry without the use of CAD/CAM | | |
| 5. Search method: | – Employed a free-text protocol with combinations of the specified keywords (and MeSH) | | |
| | – Conducted a thorough search within article references and of recently advertised dental AI applications from various manufacturers | | |

**Selection process:**
(The screening software, Rayyan, developed by the Qatar Computing Research Institute (QCRI), was used in the screening and deduplication process.) Two reviewers conducted the screening, and conflict was resolved by discussion.

| 1. Initial screening: | – Titles and abstracts of the retrieved articles were screened to assess relevance to the survey objective. |
| --- | --- |
| 2. Full-Text review: | – Articles passing the initial screening underwent a full-text review for eligibility. |

| Table 1 (continued) | |
| --- | --- |
| Data extraction: | – Relevant data from the selected articles were extracted, including details on dental CAD/CAM and AI advancements. |
| Reporting: | – The results of the search were reported in a narrative format without any quantitative data synthesis. |
| Total number of retrieved abstracts after duplicates removal before abstract screening and full-text review: | 395 |
| Articles included in this narrative literature review following inclusion and exclusion criteria and relevance: | 61 articles relative to : Artificial Intelligence integration into restorative dental CAD/CAM systems and AI applications for dental CAD/CAM materials. |

make decisions across multiple fields. The goal of super or genius AI is to achieve a higher level of intelligence than humans (*Fang et al., 2021*; *Thurzo et al., 2022a*, *2022b*). Figure 1 shows the three types of AI, their different areas, and their integration with data science.

Machine learning is a branch of AI that utilizes algorithms and statistical models to perform tasks without the need for prior knowledge or rules (*Carleo et al., 2019*). It is gaining popularity in healthcare, including dentistry (*Alqutaibi & Aboalrejal, 2023*; *Deng et al., 2023*; *Khan, Luo & Wu, 2022*; *Maruta et al., 2023*; *Patil et al., 2022*; *Prasad et al., 2023*; *Reyes et al., 2022*; *Wang et al., 2023a*; *Zhang et al., 2023b*). Two types of machine learning algorithms, supervised and unsupervised, can help healthcare providers by providing insights through data analysis (*Fatima et al., 2022*). Dentistry can benefit from machine learning in diagnosis, treatment, and automating workflows (*Fatima et al., 2022*). It has the potential to transform healthcare delivery through clinical applications. AI's fuzzy logic applications handle imprecise input data effectively in complex systems by considering natural ambiguity in classes and concepts (*Carrillo-Perez et al., 2022*). Advanced AI technologies such as cone beam computer tomography (CBCT) and 3D convolutional neural networks (CNN) are highly effective for intricate dental procedures and are also utilized in forensic dentistry (*Thurzo et al., 2022b*). Deep learning, a subfield of ML, uses multiple layers of non-linear units to analyze data and extract valuable insights. It has the potential to revolutionize the field of dentistry with impactful applications (*Aijaz et al., 2022*; *Fatima et al., 2023*; *Oztekin et al., 2023*; *Qayyum et al., 2023*). Artificial neural networks (ANN) mimic human brain processes to solve complex problems (*Basheer & Hajmeer, 2000*). ANNs are powerful tools with many medical applications, including disease diagnosis and data analysis. Their ability to learn, process information, and tolerate noise makes them very promising (*Fatima et al., 2022*). The ANN's clinical decision-making support offers healthcare professionals expert guidance to improve dental health outcomes by solving complex problems (*Ding et al., 2023b*; *Qu et al., 2022*).

AI is becoming increasingly popular as it helps reduce potential human errors, emotions, and biases, making it a perfect solution for high-risk, labor-intensive work

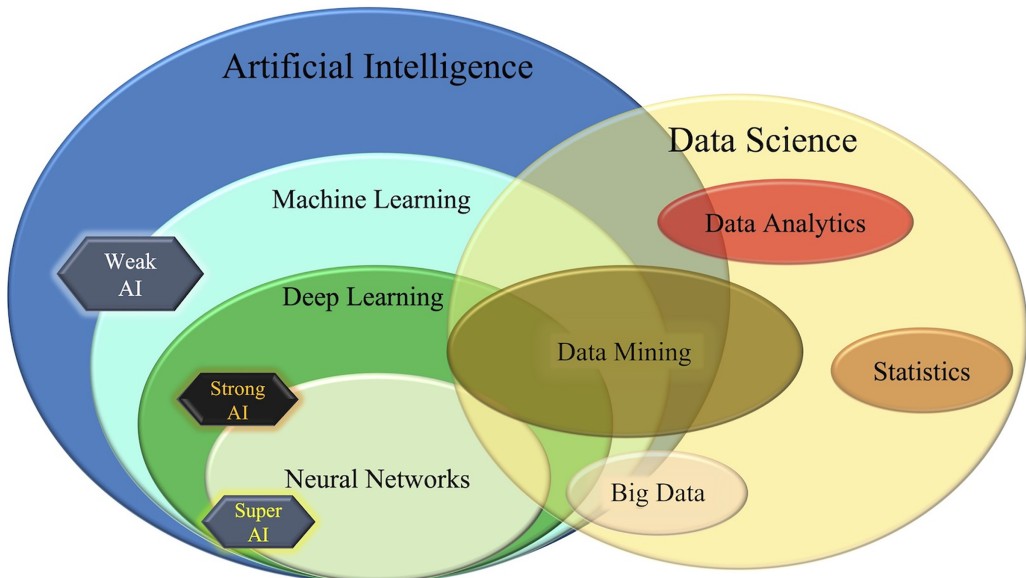

**Figure 1 The different areas of artificial intelligence (AI) and data science, and the three stages of AI.**

(*Rekow, 2020*). Dentronics refers to medical specialized robot systems and AI technology. Additionally, robotic assistants can perform their workflows repetitively without limitations (*Mayta-Tovalino et al., 2023*). AI eliminates other potential human symptoms, such as fatigue, tiredness, and boredom, often arising from repetitive work (*Joda et al., 2018*; *Rekow, 2020*). There are, however, concerns about AI's legal accountability for incorrect medical interventions due to biases in the algorithms (*Obermeyer et al., 2019*; *Thurzo et al., 2022b*). Deep AI's black-box nature can cause uncertainty, but scientists are working on "explainable AI" to clarify the decision-making process (*Ma et al., 2022*; *Oztekin et al., 2023*; *Yang et al., 2022b*; *Zhang, Weng & Lund, 2022*). Additionally, there are ethical and development controversies surrounding the creation of strong and super AI, and no applications of these types currently exist (*Ding et al., 2023b*). ML technology in dentistry is not meant to replace dentists, but rather to provide a second diagnosis based on mathematical predictions (*Mayta-Tovalino et al., 2023*). A thorough examination of the decision-making approaches and frameworks is necessary to offer definitive solutions with clear and concise explanations (*Yang, Ye & Xia, 2022a*; *Yang et al., 2021*).

## Dental computer-aided design/computer-aided manufacturing

Fabrication techniques and materials for dental restorations have advanced significantly over time. CAD/CAM systems were introduced as two-dimensional design tools by François Duret and colleagues in the early 1970s (*Duret, 1991*; *Spitznagel, Boldt & Gierthmuehlen, 2018*). CAD/CAM systems have evolved tremendously by utilizing an efficient digital workflow that eventually replaces traditional labor-intensive and time-consuming conventional fabrication of indirect dental restorations (*Ng & Poynting, 1995*; *World Dental Federation FDI, 2018*).

Dental CAD/CAM systems depend on three essential components; a scanner (computer-aided inspection (CAI)), a digital design program, and a manufacturing device (*Uzun, 2008*). Scanners digitize spatial data by capturing tooth preparation data, surrounding hard and soft tissues, and occlusion, either directly in the oral cavity or indirectly by scanning a conventional impression or poured cast (*Susic, Travar & Susic, 2017*; *Watanabe, Fellows & An, 2022*). Direct scanning, however, has a higher risk of error in cases of full arch or large area scanning due to the need for multiple images (*Güth et al., 2017*). There are three distinct types of CAD/CAM production systems for dental restorations. The first is the chair-side system, which utilizes advanced technology such as the Cerec System (Dentsply Sirona, Charlotte, NC, USA) to create and place the restoration in a single appointment. The second is the CAD/CAM dental laboratory system, like the Cerec in lab MC XL (Dentsply Sirona, Charlotte, NC, USA), which produces full restorations or copings from using and indirect scan of the impression, a stone cast, or die of the prepared tooth in the dental lab. Esthetic veneers are then added to the copings by a skilled technician. Lastly, CAD/CAM central machining networks that outsource dental lab work online. These networks are advantageous for designing and creating high-strength ceramic frameworks that have highly specialized requirements (*Alghazzawi, 2016*; *Jain et al., 2016*; *Susic, Travar & Susic, 2017*). Digital fabrication of dental restorations can either be subtractive (milling) or additive (3D printing). The use of additive manufacturing for resin-based restorations has become increasingly popular in clinical practice in recent years. However, additive manufacturing of ceramic dental restorations, such as fused deposition printed zirconia, is still developing (*Hajjaj et al., 2024*; *Wang et al., 2023b*). Many chairside and laboratory CAD/CAM systems use subtractive fabrication techniques to produce dental restorations (*Watanabe, Fellows & An, 2022*). They reduce the restoration cost due to the use of fewer materials, time, and human resources to fabricate indirect dental restorations (*World Dental Federation FDI, 2018*). Reducing material loss compared to conventional fabrication techniques makes CAD/CAM an environmentally friendly technology (*Watanabe, Fellows & An, 2022*; *World Dental Federation FDI, 2018*). Milling machines are often categorized by the number of axes they have, with three, four, or five being the most popular. For creating indirect dental restorations, machines with more axes are beneficial as they improve production efficiency. Although a five-axis machine is more costly than a three-axis one, it can complete milling tasks at a quicker pace. Overall, the milling procedure is critical for ensuring the durability of dentures and reducing construction errors (*Abduo, Lyons & Bennamoun, 2014*). Although CAD/CAM technology has its benefits, it is important to note that it does have some drawbacks, such as waste materials and milling burs wearing. This has led to a focus on improving additive manufacturing and 3D printing technologies, which have proven to be more efficient in minimizing raw materials that have been thrown away or unused (*Abduo, Lyons & Bennamoun, 2014*; *Alzahrani et al., 2023*).

Current CAD programs propose restorative designs on virtual articulators based on the scanned preparation and surrounding structures (*Watanabe, Fellows & An, 2022*). However, this is sometimes limited by the need for more intuitive digital smile design technologies and extensive occlusal data to be analyzed for many teeth (*Coachman et al.,*

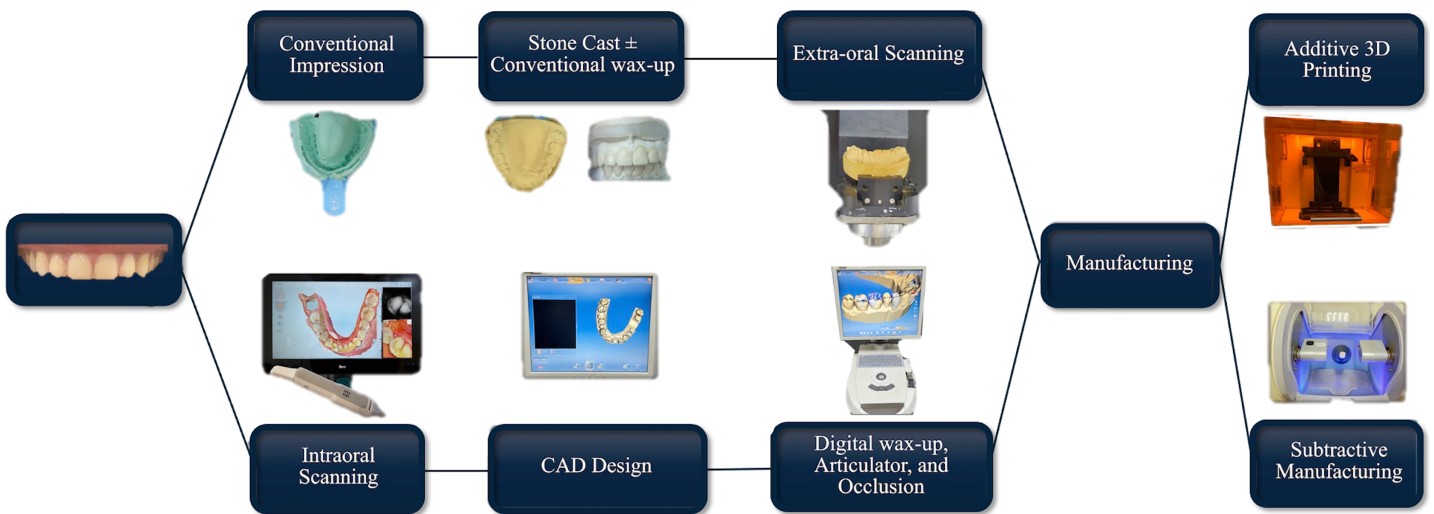

**Figure 2** Dental computer-aided design/computer-aided manufacturing (CAD/CAM) digital workflow in restorative dentistry.

*2020*). Many current CAD/CAM systems utilize user-friendly mathematically simulated articulators rather than fully adjustable articulators indicated for complex cases (*Coachman et al., 2020*; *Koralakunte & Aljanakh, 2014*; *Lepidi et al., 2021*). However, mechanical articulators are still widely used in dental labs, and they provide technicians with a real-life occlusal simulation of the clinical case at hand (*Luthra et al., 2015*). CAM fabrication can be performed with chair-side or laboratory milling machines (*Watanabe, Fellows & An, 2022*). Chair-side Economical Restoration of Esthetic Ceramic (CEREC) system is one of the most popular, easy-to-learn CAM systems since its introduction in 1985 (*Abdullah et al., 2018*; *Watanabe, Fellows & An, 2022*). Laboratory-based CAM fabrication of dental restorations can be done either in the lab or in a milling center based on the digital design sent by the lab technician (*Uzun, 2008*; *Watanabe, Fellows & An, 2022*). Currently available CAD/CAM systems are utilized worldwide and vary tremendously in regard to their potential, advantages, and disadvantages. Figure 2 demonstrates the CAD/CAM restorative digital workflow.

Dental CAD/CAM technology is also used to produce maxillofacial prostheses (*Susic, Travar & Susic, 2017*). This technology has made it possible to digitally design these prostheses. To create maxillofacial prostheses using CAD/CAM technology, the usual workflow starts with MRI and CT scans that capture both the features of the patient's soft and hard structures (*Feng et al., 2010*; *Jiao et al., 2004*). The process of developing a quick prototype model involves utilizing computer software like Materialise Mimics in Leuven, Belgium to translate data and create a wax or resin cast (*Singi et al., 2022*). This cast is then tested on the patient and any finishing touches are added by hand. The final product can be CAD/CAM fabricated rapidly from silicone elastomeric materials using a digital library that has been saved (*Singi et al., 2022*).

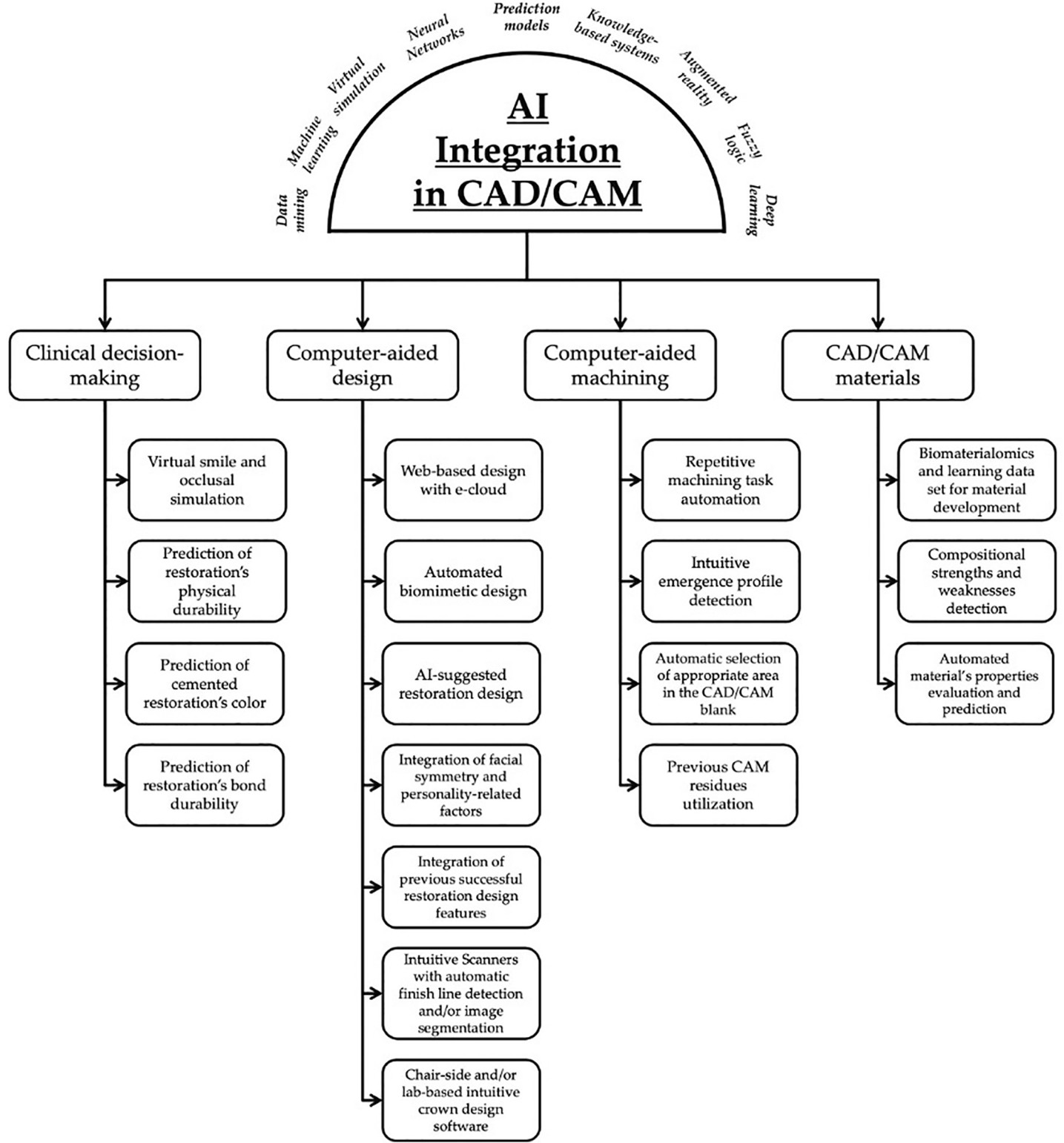

**Figure 3** Summary of artificial intelligence (AI) branches and their currently available CAD/CAM dentistry applications and enhancements.

## Artificial Intelligence integration into restorative dental CAD/CAM systems

Restorative dentistry has seen a shift in the way it operates with the introduction of artificial intelligence-based systems. This new technology has been used for automated diagnostics, prediction, and identification purposes (*Thurzo et al., 2022b*). Figure 3 summarizes some of the various AI branches and their currently available CAD/CAM dentistry applications and enhancements.

Artificial intelligence technologies and applications are gaining great interest and are utilized to expedite radiological dental diagnosis, restorative treatment planning, and prognosis prediction (*Alqutaibi & Aboalrejal, 2023*; *Schwendicke et al., 2022a*). Dental professionals have acknowledged the positive impact of AI on dental laboratories, including improved efficiency and workflow. However, they have expressed concerns about job security, professional identity, ethics, and the need for adequate training and support (*Lin et al., 2023*). AI in dentistry uses numerical reasoning, virtual simulation, ML algorithms, and deep modeling to mimic human brain function (*Naidu & Jaju, 2022*). Working with 3D models for machine learning tasks requires a lot of computational power. However, current technology is not as advanced as it is for 2D tasks, making it difficult to directly classify or regress 3D data. To reduce computation load, 3D models are often represented through sampling or depth maps, sacrificing some detail in the process (*Ding et al., 2023b*). In restorative dentistry, artificial neural networks (ANN) and convolutional neural networks (CNN) can be used for caries and/or restorations and propose appropriate excavation and restoration methods (*Mushtaq, Azam & Baig, 2022*). In the field of restorative dentistry, the digital workflows have shifted from using analog impressions to starting with an intraoral scan. New intelligent CAD/CAM systems incorporate knowledge, and some utilize machine learning in design and manufacturing. Knowledge-based systems can be considered intelligent (*Ng & Poynting, 1995*; *Upmanyu et al., 2022*). Introducing new technologies in CAD/CAM restorative dentistry increases patient satisfaction by improving the provision of dental services (*Susic, Travar & Susic, 2017*). AI utilization in CAD/CAM, such as ICAD systems (iCAD Dental, Ottowa, Canada), can result in the reduction of restorative fabrication time and an increase in design analysis and machining flexibility, benefiting clinicians and technicians (*Regassa Hunde & Debebe Woldeyohannes, 2022*). Through machine learning, millions of natural crowns can be utilized to develop optimal crown designs for specific situations (*Leeson, 2020*).

Interpretable and traceable AI technologies can be useful adjuncts for proper clinical decision-making with high accountability (*Barredo Arrieta et al., 2020*; *Li et al., 2022*). Although some dental work may result in an impressive esthetic outcome, some patients may still feel dissatisfied if the smile design does not match their personality. This can be challenging to identify whether using analog or digital methods. The design of a new smile is the most crucial and creative aspect of the digital workflow process (*Gurel, 2020*). Over the last decade, a major breakthrough has been achieved in the field of aesthetic standards and principles that hold a universal acceptance. These are grounded on the assessment of

inherent aesthetic elements, facial features, and anatomy (*Blatz et al., 2019*). Several AI-generated digital smile simulation programs and applications are available to dentists in CAD/CAM and conventional restorative dentistry. These programs, such as the Visagismile OOD (REBEL) system, can incorporate the patient's personality and facial preferences into the final smile design (*Gurel, 2020*). The system utilizes AI technology to analyze the patient's facial features and personality, followed by the application of algorithms to identify the most suitable smile design. This includes evaluating factors such as the position and shape of the front teeth, as well as the overall harmony of the individual tooth shapes. The process involves evaluating thousands of potential combinations to achieve the optimal outcome (*Alikhasi, Yousefi & Afrashtehfar, 2022*; *Gurel, 2020*). Restorative CAD/CAM dentistry can benefit from the use of Dentronics, which can improve reliability, reproducibility, accuracy, and efficiency. This can be achieved through the use of medical robot systems equipped with specialized artificial intelligence capabilities (*Grischke et al., 2020*).

When creating a CAD/CAM dental restoration, contour, and shape adjustments are typically made during the design and fabrication process. However, the evaluation of the emergence angle (EA) is often subjective (*Wang et al., 2020*). There are various methods and variations to consider when measuring the angle at which natural teeth emerge. To address this, *Saleh et al. (2022)* have proposed a new classification system and developed a machine-learning method that performs well. This approach can be helpful in designing prosthetics that closely resemble natural teeth. Additionally, the accuracy of the prediction model is influenced by the amount of data available, so increasing the amount of data can improve its performance. Overall, using this reproducible method can aid in the clinical implementation of emergence angle measurement in prosthetic design (*Saleh et al., 2022*).

Both subtractive manufacturing (SM) using CAD/CAM and 3-dimensional (3D) additive manufacturing (AM) techniques allow the incorporation of a large amount of patient data into dental restorative designs using AI (*Oldhoff et al., 2021*). In dentistry and dental practice, virtual reality (VR) is an important computer technology that's becoming increasingly essential. Using VR, dentists can design esthetically pleasing functional restorations and dentofacial prosthetic surgical guides (*Farook et al., 2020*; *Naidu & Jaju, 2022*). The use of VR in CAD/CAM restorative dentistry is gaining immense popularity, particularly with the virtual articulator application. This technology offers limitless possibilities for improving upon the mechanical articulator. It enables the analysis of both static and dynamic occlusion, as well as jaw relation, by simulating patient data with unparalleled accuracy (*Koralakunte & Aljanakh, 2014*). AI-assisted 3D virtual articulator software such as Intellifit™ TE (SensABLE Dental Technologies, Wilmington, MA, USA) includes collision detection. It allows lab technicians to touch teeth and flexibly design the restoration with more choices (*Luthra et al., 2015*). These 3D virtual articulator systems have the advantage of analyzing not only mandibular movements but also masticatory movements, such as force and frequency of contacts over time (*Koralakunte & Aljanakh, 2014*). VR facilitates highly accurate superimposition of digital impressions over 2D facial photos and/or 3D facial scans transferred to 3D models (*Coachman et al., 2020*). AI

algorithms can facilitate the detection of occlusal contacts in sequence and optimize occlusal pressure distribution on the restoration (*Revilla-León et al., 2023*). Some chairside scanner manufacturers use AI technology to reduce digital impression discrepancies caused by unwanted soft tissues, such as Itero Element 5D (Align Technology, Inc, San Jose, CA, USA) that has integrated both the AI-digital scan and design improvements as well as caries lesion detection using near infra red irradiation (*Brown, 2019*; *Moran et al., 2022*; *Schlenz et al., 2022*). Recently, ML has been successfully used to design crowns in chairside dental CAD software programs and digital scanners. Some software uses augmented reality, such as Exocad (Exocad GmbH, Darmstadt, Germany) and Ivoclar Vivadent's IvoSmile (Ivoclar Vivadent AG, Schaan, Lichtenstein), whereas 3Shape Automate (3ShapeInc, Madrid, Spain) utilizes a combination of AI, ML, and deep learning neural networks to design crowns (*Tran, 2022*). Intraoral scanners can identify implant locations and import them into AI-supported CAD software in implant dentistry to optimize the dental implant design, with later minor adjustments (*Hashem, Mohammed & Youssef, 2020*; *Roongruangsilp & Khongkhunthian, 2022*; *Schwendicke, Samek & Krois, 2020*; *Zhang et al., 2023a*). An AI algorithm can use two high-resolution intraoral scans to create a calibrated digital best-fit model for designing a passive dental prosthesis. AI-designed teeth replicate human teeth with acceptable accuracy but still require some degree of human input (*Revilla-León et al., 2021*). Recently, *Mangano et al. (2023)* demonstrated that the combination of CAD and augmented reality (AR) could revolutionize modern guided (*Giglio, Giglio & Tarnow, 2024*) placement. With this technique, clinicians can plan the implant restoration and surgery three-dimensionally using holograms instead of relying on specialized radiological methods. This approach has the potential to be both efficient and effective.

The most recent intelligent computer-aided design/computer-assisted manufacturing system was introduced in 2022. The new chair-side system (3Dme Crown automatic crown design module; Imagoworks Inc., Seoul, Republic of Korea) is web-based, using AI and cloud technologies to identify the crown preparation margins and proper design of the oral environment in seconds (*Imagoworks, 2022*). There are various reasons why dental prostheses may become dislodged, including incorrect positioning, problems with cementation, and adjustments made to the abutment's occlusal or interproximal regions. Creating a carefully crafted marginal line that follows the shape of the teeth can ensure that the restoration stays firmly in position and aids in the health of the surrounding gum tissues (*Pareek & Kaushik, 2022*). In 2020, *Lerner et al. (2020)* proposed an AI-based solution that effectively identifies subgingival margins of abutments, minimizing any possible errors. Furthermore, the solution enhances the dentist's attention to detail, particularly in maintaining interproximal and occlusal contacts while preparing the tooth (*Lerner et al., 2020*). *Cho et al. (2024)* recently utilized two deep learning methods to fully automate the design of posterior crowns. The crowns designed using deep learning were found to be comparable to those designed by technicians in terms of occlusal scheme, internal fit, tooth morphology, and proximal contacts. This could potentially reduce the need for additional design optimization by dental technicians (*Cho et al., 2024*). *Liu, Lin & Lee (2024)* showed better fit and smaller marginal gaps with digitally designed and

AI-designed crowns. Technicians' designs were used by *Tian et al. (2021*, *2022)* to train 2-stage generative adversarial network (GAN) models in generating crowns. The training data consisted of 2D depth maps obtained from real-world patient-database-obtained 3D tooth models (*Tian et al., 2021*). The first stage of the proposed model involved a conditional GAN (CGAN) that learns how to restore the occlusal relationship between a defective tooth and its definitive restoration. In the second stage, an improved CGAN was created by adding an occlusal groove parsing network (GroNet) and an occlusal fingerprint constraint to enhance the functional anatomical form of the occlusal surface (*Tian et al., 2021*, *2022)*. This model outperformed sophisticated DL models for occlusal surface reconstruction (*Tian et al., 2022*). In 2023, *Chau et al. (2023)* and *Ding et al. (2023a)* demonstrated that the 3D GAN and 3D-DCGAN networks are beneficial in automatically designing biomimetic dental restorations by learning the occlusal anatomic features of similar remaining teeth. *Farook et al. (2023)* created a new 3D dental prosthetic dataset for 3D convolutional neural network (3D-CNN) training purposes, as virtual data were successfully used in both open-source and commercial CAD workflows to create tooth preparations for machine learning (ML) in restorative dentistry. 3D deep learning (DL) was able to accurately produce inlays and onlays for suitable tooth preparations (*Farook et al., 2023*).

AI has also been used to improve the flexibility, safety, efficiency, and intuitiveness of the CAM part of the dental CAD/CAM restoration fabrication process. These systems, such as the AI DRIVEN (CIM system, Milan, Italy), *CIMsystem (2021)* can automate repetitive digital workflow steps, such as preparation margin detection and/or undercut designation. Similar to AI in CAD, AI in CAM will eventually develop into daily efficient digital dentistry if implemented accurately. Maxillofacial prosthesis can benefit from AI-assisted CAD/CAM technologies like the bionic eye. With a glasses-mounted smart camera, patients can see and recognize faces or understand text. AI expert analyzes the camera's data and converts it into audio, which is then provided to visually impaired individuals (*Bernauer, Zitzmann & Joda, 2021*).

## AI applications for dental CAD/CAM materials

Recently, several studies have investigated the usefulness of AI for evaluating the prognosis of dental restorations and their materials. *Yamaguchi et al. (2019)* found that convolutional neural networks (CNN) can be particularly useful in determining the prognosis of CAD/CAM resin crown bond longevity using 3-dimensional stereolithography models of a scanned die. *Kose et al. (2023)* developed an ML decision tree regression model to predict the final color of cemented veneers. Clinicians can use this method to predict the ultimate color of ceramic veneers made from leucite-reinforced glass CAD/CAM ceramic high translucency and low translucency blanks following their cementation with translucent types of cement. The prediction is based on the thicknesses of the ceramic and the substrate's color (*Kose et al., 2023*).

Fuzzy logic has traditionally been applied in color and optics research and applications (*Gur, Mendlovic & Zalevsky, 1999*). In the last decade, there have been proposals to use it to evaluate changes in dental and dental CAD/CAM materials' color (*Herrera et al., 2010*)

Traditionally, numerical methods have been used to perform optimization for optoelectronic inference fuzzy logic systems. However, this area of study is still actively being researched (*Herrera et al., 2010*).

Lately, researchers have been combining artificial intelligence, particularly machine learning, with traditional experimental methods to discover connections between the properties of restorative materials and various multifaceted physical factors (*Butler et al., 2018*; *Schwendicke, Samek & Krois, 2020*). ML is a faster method compared to traditional experimental methods because it can quickly assess and analyze collected data, extracting relevant features, and saving time (*Chen & Gu, 2019*). For effective machine learning prediction of CAD/CAM raw materials properties abundant training data, descriptors, and appropriate algorithms are needed (*Ward et al., 2016*). *Li et al. (2022)* utilized machine learning to successfully evaluate the flexural strength (FS) of several resin composite CAD/CAM blocks and identified the fillers (mainly $SiO_2$, barium glass, and $Al_2O_2$) and monomers (mainly urethane dimethacrylate (UDMA)) that could improve the FS of resin composite CAD/CAM materials. However, the data used for the ML models were inadequate. They required extrapolations beyond the training sets, which was evident in the uncertainties provided by the random forest and extreme gradient boosting models (*Li et al., 2022*). To improve the survival rate of CAD/CAM resin composite (CR) crowns, it is highly desirable to establish a protective measure against the restoration's debonding. In this study, a CNN method was used for deep learning which showed great success in predicting the likelihood of debonding in CAD/CAM resin composite crowns using 3D stereolithography models of a patient's die (*Yamaguchi et al., 2019*). Recently, *Grymak et al. (2023)* determined that the long short-term memory (LSTM) neural network model, created with Python, has the potential to decrease the simulation duration and the number of specimens needed for wear testing different dental materials. This could also possibly enhance the accuracy and dependability of wear testing forecasts (*Grymak et al., 2023*).

According to data mining research, changing the composition of restorative material affects the durability of intraoral dental restorations (*Babu, Onesimu & Sagayam, 2021*). The process-structure-property (PSP) framework for materials suggests that the history of the manufacturing process, hierarchical structure, and material properties are all causally related. Biomaterialomics extends these concepts to address scientific and technological challenges in developing biomaterials and biomedical devices (*Basu et al., 2022*). Integrating AI concepts and data algorithms into biomaterials science could benefit the future of dental CAD/CAM materials and facilitate the fabrication of personalized dental implants and 3D-printed restorations using AM (*Basu et al., 2022*). The availability of sufficient and accurate data is essential for successfully utilizing AI in dental CAD/CAM practices. Thus, registration of authentic data by dentists, technicians, and dental researchers could expedite the complete incorporation of AI into CAD/CAM dentistry and dental materials (*Singi et al., 2022*). For the non-surgical digital restoration of facial-dental deformed dermal structures, researchers at the Federal Polytechnic School of Zurich, Switzerland and the California Institute of Technology in Pasadena, USA have developed artificial skin using AI technology. This maxillofacial prosthetic CAD/CAM material can detect 5 °C to 50 °C temperature fluctuations (*Bernauer, Zitzmann & Joda, 2021*).

While AI cannot entirely replace the dentist and dental material researcher, *Naidu & Jaju (2022)* exploring the effect of various factors on new CAD/CAM materials would benefit building the learning data sets needed and expedite the utilization of AI technology enhancements in the processes of digital restorative dentistry.

## CONCLUSIONS

In conclusion, AI in CAD/CAM restorative dentistry uses numerical reasoning, virtual simulation, machine learning algorithms, and deep modeling to replicate human brain function. These technologies can improve patient satisfaction, reduce restorative fabrication time, and enhance design analysis and machining flexibility. Interpretable and traceable AI technologies can aid in clinical decision-making with high accountability. Virtual reality and machine learning AI applications further enhance breakthroughs in aesthetic CAD/CAM restorative dentistry. AI branches are being integrated into CAD/CAM dentistry. Technologies like convolutional neural networks and machine learning decision tree models show promise in evaluating dental restorations. Fuzzy logic may be used to assess changes in dental materials' color. These technological advancements can enhance dental treatments. More research is needed with the use of neural networks in dentistry to put them into daily practice and to facilitate the work of dentists.

### Funding

This research work was funded by Institutional Fund Projects under grant no. (IFPRP: 609-165-1442) and received technical and financial support from the Ministry of Education and King Abdulaziz University, DSR, Jeddah, Saudi Arabia. The funders had no role in study design, data collection and analysis, decision to publish, or preparation of the manuscript.

### Grant Disclosures

The following grant information was disclosed by the authors:
Institutional Fund: IFPRP: 609-165-1442.
Ministry of Education and King Abdulaziz University, DSR, Jeddah, Saudi Arabia.

### Competing Interests

The authors declare that they have no competing interests.

### Author Contributions

- Hanin E. Yeslam conceived and designed the experiments, performed the experiments, analyzed the data, prepared figures and/or tables, authored or reviewed drafts of the article, and approved the final draft.
- Nadine Freifrau von Maltzahn conceived and designed the experiments, performed the experiments, analyzed the data, authored or reviewed drafts of the article, and approved the final draft.

- Hani M. Nassar conceived and designed the experiments, performed the experiments, analyzed the data, prepared figures and/or tables, authored or reviewed drafts of the article, and approved the final draft.

## Data Availability

This is a literature review.

## Supplemental Information

Supplemental information for this article can be found online at http://dx.doi.org/10.7717/peerj.17793#supplemental-information.

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
