# Peer review of "Revolutionizing CAD/CAM-based restorative dental processes and materials with artificial intelligence: a concise narrative review"

_PeerJ, doi:10.7717/peerj.17793_

## Round 0.1 · original submission · Major Revisions

I trust this message finds you well. I have had the opportunity to review the manuscript titled "Revolutionizing CAD/CAM-based dental processes and materials with artificial intelligence: a concise overview," which you submitted to PeerJ Journal. While the article's topic is of significant interest and has the potential to make a valuable contribution to the field, there are some critical points that need to be addressed to enhance the overall quality and clarity of the manuscript.

There is repetitive information regarding the language of the included articles and how they were analyzed and vital information is missing from the methodology, such as the inclusion criteria, the number of included articles, and the specifics of the review process, including the number of reviewers and the software used for data collection.

Thank you for your attention to these matters, and I look forward to reviewing the revised version of your article.

Sincerely,

Reviewer 1 ·

Basic reporting

The manuscript “Revolutionizing CAD/CAM-based dental processes and materials with artiûcial intelligence: a concise overview” with the aim of review the AI possibilities inside CAD-CAM process has an interesting purpose. Despite that, it presents some issues that need to be addressed.
- Please remove the section inside the introduction, present a continuous text.
- If only restorative dentistry field articles were considered, this need to be specified on the article’s title and abstract.
- As a suggestion, I would put the “Artificial Intelligence (AI) and its pertinent branches” section before the section “Dental computer-aided design/computer-aided manufacturing (CAD/CAM)”.

Experimental design

- There are some repetitive information in the methodology section, such as the language of the included articles and how they were analyzed. Please, review the writing.
- There are some information missing for the methodology, also. As the inclusion criteria, number of included articles, and how they were reviewed (number of reviewers, what software were used to collect the data).
- Figure 1 could be more illustrative and attractive.

Validity of the findings

- Some statements could be more carefully addressed considering that some of the AI technologies that are mentioned in the manuscript have only been assessed by a low number of papers or even not fully tested/validated in a clinical scenario. Therefore, sentences as “This approach is efficient and effective” (line 335) should be reviewed.

·

Basic reporting

First I would like to congratulate the authors for a well presented article. The paper brings an actual theme with an interesting approach. Before acceptance, I would like to recommend some minor revisions.

The abstract introduction can be more concise. The authors may present some AI applications in the abstract, giving examples of how this technology can be associated with CAD-CAM procedures.

The Introduction's two final sentences have the same idea as the Relevance and Target Audience section that comes next. I kindly suggest reorganizing the ideas.

Experimental design

No comment.

Validity of the findings

In the Dental computer-aided design/computer-aided manufacturing (CAD/CAM) section, the authors discuss about additive manufacturing. Please, make it clear that this approach is not actually used for ceramic restorations.

In line 302, there is a abbreviation for additive manufacturing (AM), but not to subtractive, please provide the abbreviation.
In line 304, VR is used without the written name, which is stated in the next line.

The conclusion is too long. Please be more concise.

Additional comments

Some cited references along the text are not properly formatted. Please, when referring to various authors in one study, use "et al.".

Please, use the term resin composite instead of composite resin.

Reviewer 3 ·

Basic reporting

The manuscript is well-written, clear and adequate for new professionals who want to learn more about AI and CAD-CAM system. However, there are some major points to be reviewed before a new consideration for publication. The added figures must be reviewed.

Experimental design

Abstract:
1- The abstract is well-written. However, it lacks more precise information. Authors should state some specific applications of the AI during the clinical procedures.
Methods
2- “The results were reported in a narrative format, without any quantitative data synthesis. The search only included articles written in English. The results were reported in a narrative format, without any quantitative data synthesis.” These information were already cited in the same paragraph. Please, review it.
3- The authors should add the precise search strategy applied for each database, in order to allow the reproducibility of the applied method.
4- The quality of the added images in the figure 1 are poor. I suggest the replacement of they.
5- In fig.2, the second level of AI is difficult to read. Please change the color and review the quality of all figures.

Validity of the findings

As a narratie review, the study was successful in addressing the use of artificial intelligence within dentistry together with the cad-cam system, with a simple search strategy.
The conclusions are well stated and meet the objective of the suty.

Additional comments

No additional comments.

---

## Round 0.2 · accepted · Accept

Dear author,

Thank you for submitting your manuscript titled “Revolutionizing CAD/CAM-based restorative dental processes and materials with artificial intelligence: a concise narrative review” to PeerJ. I appreciate the effort you and your co-authors have put into addressing the reviewers’ comments. After carefully reviewing the revised version against the original reviews, I am pleased to inform you that your manuscript has been accepted for publication.